# Prediction of Cervical Cancer Outcome by Identifying and Validating a NAD+ Metabolism-Derived Gene Signature

**DOI:** 10.3390/jpm12122031

**Published:** 2022-12-08

**Authors:** Aozheng Chen, Wanling Jing, Jin Qiu, Runjie Zhang

**Affiliations:** Obstetrics and Gynecology Department, Tongren Hospital, Shanghai Jiao Tong University School of Medicine, 1111 XianXia Road, Shanghai 200336, China

**Keywords:** NAD+, NAD+ metabolic-related genes, cervical cancer, prognosis, constructed model

## Abstract

Cervical cancer (CC) is the second most common female cancer. Excellent clinical outcomes have been achieved with current screening tests and medical treatments in the early stages, while the advanced stage has a poor prognosis. Nicotinamide adenine dinucleotide (NAD+) metabolism is implicated in cancer development and has been enhanced as a new therapeutic concept for cancer treatment. This study set out to identify an NAD+ metabolic-related gene signature for the prospect of cervical cancer survival and prognosis. Tissue profiles and clinical characteristics of 293 cervical cancer patients and normal tissues were downloaded from The Cancer Genome Atlas database to obtain NAD+ metabolic-related genes. Based on the differentially expressed NAD+ metabolic-related genes, cervical cancer patients were divided into two subgroups (Clusters 1 and 2) using consensus clustering. In total, 1404 differential genes were acquired from the clinical data of these two subgroups. From the NAD+ metabolic-related genes, 21 candidate NAD+ metabolic-related genes (ADAMTS10, ANGPTL5, APCDD1L, CCDC85A, CGREF1, CHRDL2, CRP, DENND5B, EFS, FGF8, P4HA3, PCDH20, PCDHAC2, RASGRF2, S100P, SLC19A3, SLC6A14, TESC, TFPI, TNMD, ZNF229) were considered independent indicators of cervical cancer prognosis through univariate and multivariate Cox regression analyses. The 21-gene signature was significantly different between the low- and high-risk groups in the training and validation datasets. Our work revealed the promising clinical prediction value of NAD+ metabolic-related genes in cervical cancer.

## 1. Introduction

Cervical cancer is a huge burden for women around the world and a global health concern, as well as becoming one of most frequent cancers that threaten the health and lives of women. CC has contributed significant cancer-associated deaths, especially in developed countries [1,2]. Patients are treated based on the established treatment standards by the international Federation of Gynecology and Obstetrics (FIGO) and the National Comprehensive Cancer Network (NCCN). As treatment measures have matured, most cases of early disease achieve a satisfactory level of survival and prognosis. However, for the advanced stage, the unoptimistic prognosis is ascribed to a lack of timely and accurate measures depending on the clinical condition of patients [3,4,5]. Metastasis and recurrence are significant hurdles to the prognosis of cervical cancer. Hence, there is an urgent need to identify biomarkers of progression to reverse the outcome of CC.

Cervical cancer cells are involved in energy metabolism reprogramming, which promotes their rapid cell proliferation [6,7]. Malignant tumor cells display increased lactate production, which preferentially requires aerobic glycolysis over oxidative phosphorylation. NAD+ serves as a vital coenzyme for energy transduction and aerobic glycolysis, which mediates redox reactions in metabolic pathways [8], as well as being a substrate for NAD+-dependent enzymes, such as sirtuins and poly ADPribose polymerases (PARPs) [9,10]. NAD-mediated signaling events regulate transcription, cell cycle progression, DNA repair, and metabolic regulation through these enzymes [11]. NAD+ is essential for various cancer-related processes including the citric acid (TCA) cycle, oxidative phosphorylation, fatty acid metabolism, and antioxidant defense metabolism [12].

Accumulating evidence has uncovered that cancer cells exhibit a dependency on metabolic pathways regulated by NAD+. Of note, decreasing the level of NAD+ suppresses glycolytic activity and causes a metabolic collapse in numerous cancers [13,14,15]. Nicotinamide (NAM) plays an essential role in the metabolic pathway that produces NAD+ [16]. As a precursor to the coenzyme NAD+, nicotinamide participates in cellular energy metabolism in the mitochondrial electron transport chain [17]. Researchers have found that nicotinamide inhibited the proliferation of HeLa cells and significantly increased ROS accumulation and GSH depletion at relatively high concentrations [18]. Moreover, nicotinamide phosphoribosyltransferase (Nampt) catalyzes the rate-limiting step of the mammalian NAD salvage pathway. More notably, nicotinamide phosphoribosyltransferase (Nampt) levels increased with the SCC grade [19].

This study constructed and validated an NAD+ metabolic-related gene prognostic signature and explored the molecular signatures in cervical cancer. Through integrative analysis of the relationship between NAD+ metabolism and CC, this study may help in providing potential predictors and treatments of CC.

## 2. Materials and Methods

### 2.1. Dataset Processing

We downloaded the clinical characteristics and NAD+ metabolic-related genes of 309 CESC patients, including cervical squamous cell carcinoma and endocervical adenocarcinoma patients and non-cervical cancer tissues refer to adjacent normal tissues, from the TCGA portal (https://portal.gdc.cancer.gov/) in October 2021. The inclusion criteria were as follows: (a) samples diagnosed as CC; (b) samples with mapped clinical information and gene expression matrix; (c) samples with complete clinical information including age, FIGO stage, tumor size status, lymph node status and metastasis status. Samples lacking follow-up data were excluded. Patients were randomly split into a training set for identification and a testing set for validation in Table 1.

### 2.2. NAD+ Metabolic-Related Genes

NAD+ metabolic-related genes were identified through chi-square test analysis, and the *p* value < 0.05 was set up for infiltration data. All cervical cancer patients were divided into Cluster 1 or Cluster 2 via the Consensus Cluster Plus R package [20]. Survival analysis was conducted, and the expression levels between the two subgroups were assessed.

### 2.3. Construction and Validation of the NAD+ Metabolic-Related Gene Prognosis Model

Multivariate Cox regression analysis was used to identify 21 significant NAD+ metabolic-related genes for construction of the prognostic signature. The risk score for each cervical cancer patient was calculated as per the following formula: Risk score = Σ Expn *βn, where Expn stands for the expression value of each gene, and βn represents the coefficient of each target gene. We then divided cervical cancer patients into the low-risk or high-risk group based on the cut-off risk score.

### 2.4. Application and Assessment of Prognosis Model

In order to evaluate the use of the signature, Kaplan–Meier survival curves were applied to calculate the difference in the progression-free interval (PFI) and overall survival (OS) between the different risk groups in the training and testing sets. Time-dependent receiver operating characteristic (ROC) curves were applied to verify the prediction accuracy of this constructed prognostic model. Univariate and multivariate Cox regression analyses further validated independent clinical prognoses. Nomograms that included age, FIGO stage, tumor size, lymph node status, metastasis status, and risk score were used to predict the 5- and 10-year survival probabilities through Kendall’s tau-b correlation analysis.

### 2.5. Function Enrichment Analysis

The Gene Ontology (GO) analysis method explores the possible KEGG pathways that might implicate the gene-predicted signature. The reference gene set was retrieved from Cluster Profiler [21], and the pathways were considered significant when *p* ≤ 0.05 or FDR ≤ 0.05.

### 2.6. Statistical Analysis

All statistical analyses were carried out using R version 4.0.2. Differentially expressed genes were identified using a chi-square test. Kendall’s tau-b correlation analysis was used to estimate the correlation analysis of clinical factors and risk scores. Survival differences were assessed through Kaplan–Meier analyses and log-rank tests. A *p* value or FDR ≤ 0.05 was regarded as statistically significant.

## 3. Results

### 3.1. Clinical Data and Identification of NAD+ Metabolic-Related Genes in Cervical Cancer

The clinical characteristics and 39 NAD+ metabolic-related genes of 293 cervical cancer patients and normal tissues were confirmed. There were 10 significantly differentially expressed genes: NADSYN1, PARP9, PARP14, SIRT1, SIRT3, BST1, PTGIS, QPRT, AOX1, NT5E (Wilcoxon test *p* < 0.05; Figure 1A). To explain the influences of NAD+ metabolic-related genes in the development of CC, we divided CC samples into Clusters 1 (*n* = 148) and 2 (*n* = 145) via the Consensus Cluster Plus R package and found that k = 2 realized the optimal clustering stability from k = 2 to k = 9 (Figure 1B). Demographic, clinical, and pathologic characteristics of the patients with CC are shown in Table 1. Principal components analysis (PCA) showed significant differences between Cluster 1 and Cluster 2 subtypes (Figure 1C). The PFI and OS of Cluster 2 were shorter than those of Cluster 1 (Figure 1D,E).

### 3.2. Establishment and Validation of the NAD+ Metabolic-Related Gene Signature of CC

To establish potential NAD+ metabolic-related genes for predicting survival and prognosis, we compared differentially expressed genes between Cluster 1 and Cluster 2; in total, 1404 differentially expressed genes were acquired, of which 1064 genes were significantly upregulated, while 340 were significantly downregulated. Figure 2A,B show the volcano plots and heatmap. The ten most significant up- and downregulated correlated genes are presented in Table 2. In total, 21 NAD+ metabolic-related genes (ADAMTS10, ANGPTL5, APCDD1L, CCDC85A, CGREF1, CHRDL2, CRP, DENND5B, EFS, FGF8, P4HA3, PCDH20, PCDHAC2, RASGRF2, S100P, SLC19A3, SLC6A14, TESC, TFPI, TNMD, ZNF229) from the multivariate Cox regression were identified to construct the NAD+ metabolic-related prognostic signature (Figure 2C). The risk score = (0.00178665831343545*ADAMTS10) + (0.0728069429570205*ANGPTL5) + (0.00116045523502544*APCDD1L) + (0.0184775950770853*CCDC85A) + (0.000697519709368892*CGREF1) + (−0.00367113285128586*CHRDL2) + (0.00445754551419468*CRP) + (−0.00182672501345106*DENND5B) + (0.000255094532012467*EFS) + (0.00205557512722222*FGF8) + (0.00177634101015653*P4HA3) + (0.0243834975449747*PCDH20) + (0.00446496406641565*PCDHAC2) + (0.00151667627062981*RASGRF2) + (0.000027497581477097*S100P) + (0.000993965780981564*SLC19A3) + (0.0000598317363640983*SLC6A14) + (0.00016753983756972*TESC) + (0.000245045754598803*TFPI) +(0.0407767196909512*TNMD)+ (0.00506653984056587*ZNF229). According to the cut-off value (cut-off value = 0.000120583847922351), risk groups were divided into high- and low-risk groups in the training and validation datasets. The risk score and survival status data of these two groups are shown in Figure 2D. The heatmap depicts the expression patterns of NAD+ metabolic-related genes in the training and validation datasets (Figure 2E). The survival analysis illustrated that the high-risk group had a significantly worse PFI and OS compared with those of the low-risk group in both the training and validation cohorts (Figure 2F,H,J,L). The AUC value of the NAD+ metabolic-related gene signatures was 0.830 and 0.772 in the training set (Figure 2G,K), demonstrating the accuracy of this predictive model for CC survival. This model was then verified in the testing cohort, and the AUC was 0.803 and 0.787 (Figure 2I,M).

### 3.3. Prognostic Model Correlated with Clinicopathological Characteristics

To examine the prognostic value of our model, we used both the univariate and multivariate Cox regression analyses on CC clinical characteristics including age, FIGO stage, tumor size, lymph node status, metastasis status, and risk group, which revealed the risk group could be the independent predictive factor of CC (*p* < 0.001). All the details are shown in Table 3. Observations revealed that the PFI and OS in the low-risk group were significantly longer than those in the high-risk group in patients aged both >45 and ≤45 years; in FIGO stages I and II–IV; in tumor sizes ≤ 4, >4 cm, and Tx; and in the N0, N1, Nx, M0, M1, and Mx classifications. (Figure 3A–D,I–L, Figure 4A–C,G–I, Figure 5A–C,G–I and Figure 6A–C,G–I). ROC analyses revealed the sensitivity and specificity of the 21-NAD + metabolic-related-gene signature in predicting CC in patients aged both >45 and ≤45 years; in FIGO stages I and II–IV; in tumor sizes ≤ 4, >4 cm, and Tx; and in the N0, N1, Nx, M0, M1, and Mx classifications. The AUC values of the ROC curve for PFI and OS were 0.830 and 0.751 in patients aged ≤45 (Figure 3E,G), and 0.853 and 0.786 in patients aged >45 (Figure 3F,H). The AUC values of the ROC curve for PFI and OS were 0.854 and 0.795 in FIGO stage I (Figure 3M,O), and 0.802 and 0.742 in FIGO stage II–IV (Figure 3N,P), respectively. The AUC values of the ROC curve for PFI and OS were 0.857 and 0.847 in tumor size ≤ 4 (Figure 4D,J), 0.797 and 0.715 in tumor size > 4 cm (Figure 4E,K), and 0.847 and 0.781 in tumor size Tx (Figure 4F,L). The AUC values of the ROC curve for PFI and OS were 0.860 and 0.874 in lymph node N0 (Figure 5D,J), 0.711 and 0.591 in lymph node N1 (Figure 5E,K), and 0.855 and 0.779 in lymph node Nx (Figure 5F,L). The AUC values of the ROC curve for PFI and OS were 0.875 and 0.811 in M0 (Figure 5D,J), 0.800 and 0.900 in M1 (Figure 6E,K), and 0.805 and 0.750 in Mx (Figure 6F,L). The results of the Kaplan–Meier and ROC analyses of the different groups are shown in Table 4.

### 3.4. Prognostic Nomogram and Functional Enrichment Analysis

Prognostic nomograms incorporating clinicopathological characteristics and the prognostic signature of the 21 NAD+ metabolic-related genes were constructed to quantify the probabilities for predicting the 5- and 10-year PFI and OS of cervical cancer patients (Figure 7A,B). We further performed functional enrichment analysis of GO and KEGG and selected the top 30 enriched GO terms and KEGG pathways, which are shown in Figure 7C,D. Biological process (BP), cellular component (CC), and molecular function (MF) revealed NAD+ metabolic-related genes involved in extracellular structure organization, the extracellular matrix, the PI3K-Akt signaling pathway, and the MAPK signaling pathway, which participate in the process and development of CC.

## 4. Discussion

NAD+ is an essential coenzyme that mediates glycolysis, where increased NAD+ levels facilitate the rate of aerobic glycolysis and initiate cancer cells via lactate dehydrogenase (LDH) and glyceraldehyde 3-phosphate dehydrogenase (GAPDH) [22]. NAD+ can be synthesized via the de novo pathway, Preiss–Handler pathway, and salvage pathway from tryptophan, nicotinic acid (NA), and nicotinamide (NAM), respectively. An increase in the levels of NAD+ enhanced the malignant phenotype in the cells and led to poorer outcomes, accompanied by NAMPT overexpression, which is mainly produced by the NAD+ salvage pathway in cancer cells and involved in several pro-cancer pathways mediated by NAMPT [23,24,25]. NAMPT was observed to be overexpressed in a broad range of malignant tumors and to modulate cancer cell proliferation, metastasis, and induction of angiogenesis [13,14,26]. Consumption of NAD+ sensitizes cancer cells to oxidative damage by disrupting antioxidant defense systems, reducing cell proliferation, and triggering the cell death of manipulating cell signaling pathways [27,28]. As major NAD+ consumers, SIRTs, PARPs, and CD38 function in many critical processes in cancers. Different SIRT isoforms are essential for DNA repair, energy metabolism, and transcription in cancer cells. Several recent studies found that SIRT modulators exhibit potent biological effects in ovarian, endometrial, and breast cancer cells [29,30,31]. PARPs play pivotal roles in NAD+-dependent DNA damage repair and transcriptional regulation involved in cancer progression. PARP inhibitors have been comprehensively used for gynecologic cancers [32]. Studies demonstrate that CD38 promotes cervical cancer cell growth by inhibiting apoptosis and reducing ROS levels [33]. This necessity supports the idea that NAD+ metabolism is involved in cancer development and progression, and that targeting NAD+ metabolism is likely to be a very intriguing therapeutic concept in repressing underlying tumor progression.

In our study, we first explored clinical characteristics, confirmed 39 NAD+ metabolic-related genes among 293 cervical cancer patients and para-cancerous tissues, and identified 10 differentially expressed genes. By dividing CC samples into clusters and further investigating NAD+ metabolic-related genes implicated in CC progression, we pinpointed 21 prognostic NAD+ metabolic-related genes (ADAMTS10, ANGPTL5, APCDD1L, CCDC85A, CGREF1, CHRDL2, CRP, DENND5B, EFS, FGF8, P4HA3, PCDH20, PCDHAC2, RASGRF2, S100P, SLC19A3, SLC6A14, TESC, TFPI, TNMD, ZNF229) using univariate and multivariate regression analyses. To explore the clinical value of these 21 genes, the signature was validated in the testing and training cohorts, indicating its reliability and efficacy. The signature could effectively classify CC patients into high-risk and low-risk groups in both sets. The predicated model showed a strong predictive performance, and the survival rates of the low-risk group were significantly higher than those of the high-risk group. The AUC values of the ROC curve were 0.830 and 0.772, and 0.803 and 0.787, respectively, in the training and testing sets, which proved that this signature has reliable accuracy as a prognostic factor in CC patients. Moreover, the results of the multivariate analysis in different datasets clarified that the NAD+ metabolic-related signature is an independent factor after adjusting for other clinical factors (including age, FIGO stage, tumor size, lymph node status, metastasis status, and risk group).

Among the 21 candidate genes, ADAMTS10, ANGPTL5, APCDD1L, CCDC85A, CGREF1, CHRDL2, DENND5B, EFS, P4HA3PCD, H20, PCDHAC2, RASGRF2, SLC6A14, TESC, TNMD, and ZNF229 have not been reported in CC; our study is the first to characterize these genes as a prognostic signature. SLC19A3, CRP, and FGF8 have been reported to have clinical prognostic value in CC progression [34]; our study is the first to characterize SLC19A3 as a prognostic signature through NAD+ metabolism. TFPI has been reported to be related to survival in breast cancer [35]. S100P has been reported as a prognostic signature in cholangiocarcinoma [36] and hepatocellular carcinomas [37]. Moreover, our gene functional enrichment analysis revealed that the related genes were enriched in the process of the extracellular matrix and the PI3K-Akt and MAPK signaling pathways. Research has highlighted that the ECM initiates cell proliferation, migration, and invasion in CC [38,39]. Previous studies illustrated that the PI3K-Akt and MAPK signaling pathways are involved in the progression and metastasis of CC as well as being therapeutic targets [40,41].

In this study, we investigated the functional implication of NAD+ and genes related to NAD+ metabolism for the outcome of CC. NAD+ metabolism is involved in pro-cancer pathogenesis and is considered a promising therapeutic for oncotherapy. This NAD+ metabolic-related signature is based on population databases; further in vitro or in vivo experimental data as well as clinical experiments are needed to confirm our findings.

Recent research shows that NAD+ metabolism-derived gene signatures could predict the prognosis of ovarian cancer and immunotherapy [42]. In some ways, NAD+ metabolism-related gene signatures (models) could be associated with the prognosis in some types of carcinomas. Additionally, we also explored the characterization of 22 types of immunocyte infiltration and the proportion of immunocyte infiltration in the two NAD+ metabolic-related clusters in cervical cancer, but there were no statistically significant differences. Perhaps this difference could be attributed to the different types of tumors; thus, more experiments are needed for further exploration.

In conclusion, we established a NAD+ metabolic-related gene signature for CC. The external validation of this signature was significantly correlated to the risk value and OS of CC patients, and this signature represents a promising biomarker to predict cervical cancer prognosis. This study demonstrates a new NAD+ metabolism-related prognostic model and therapeutic target for cervical cancer.

## Figures and Tables

**Figure 1 jpm-12-02031-f001:**
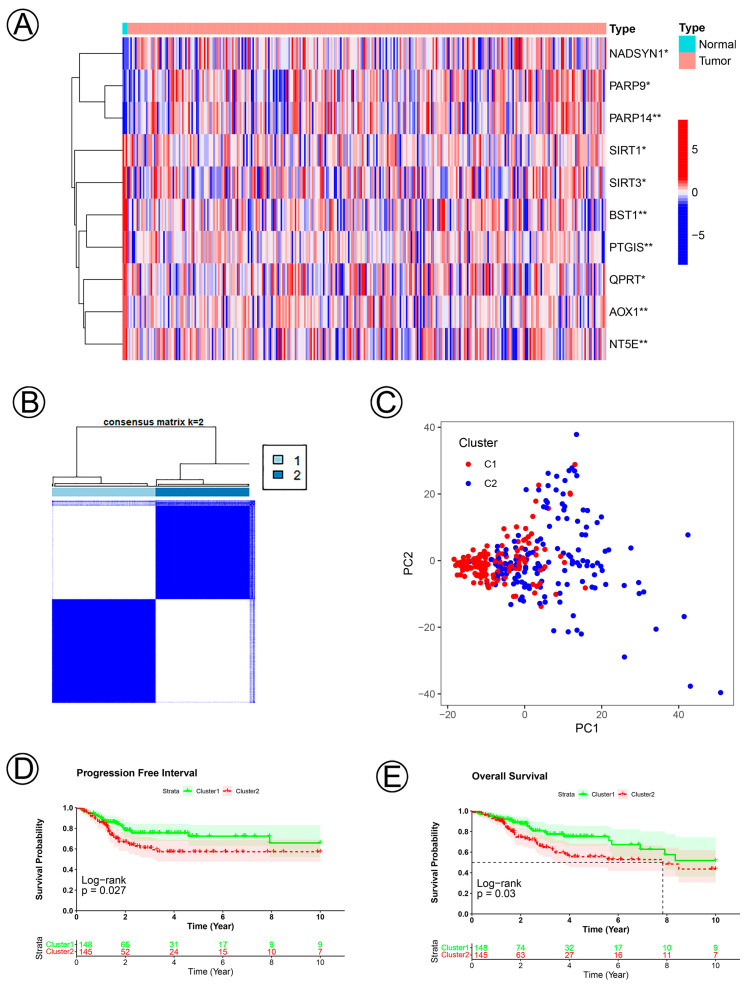
NAD+ metabolic-related clusters were determined according to 10 NAD+ metabolic-related genes in cervical cancer. (**A**) A heatmap of the expression levels of 10 NAD+ metabolic-related genes between the normal and tumor tissues. (**B**) Consensus cluster matrix of cervical cancer tumor samples when k = 2. (**C**) Two-dimensional principal component analysis for NAD+ metabolic-related clusters. The red dots represent C1, and the blue dots represent C2. (**D**,**E**) KM curves of PFI and OS for the patients with cervical cancer grouped by Cluster 1 and Cluster 2.

**Figure 2 jpm-12-02031-f002:**
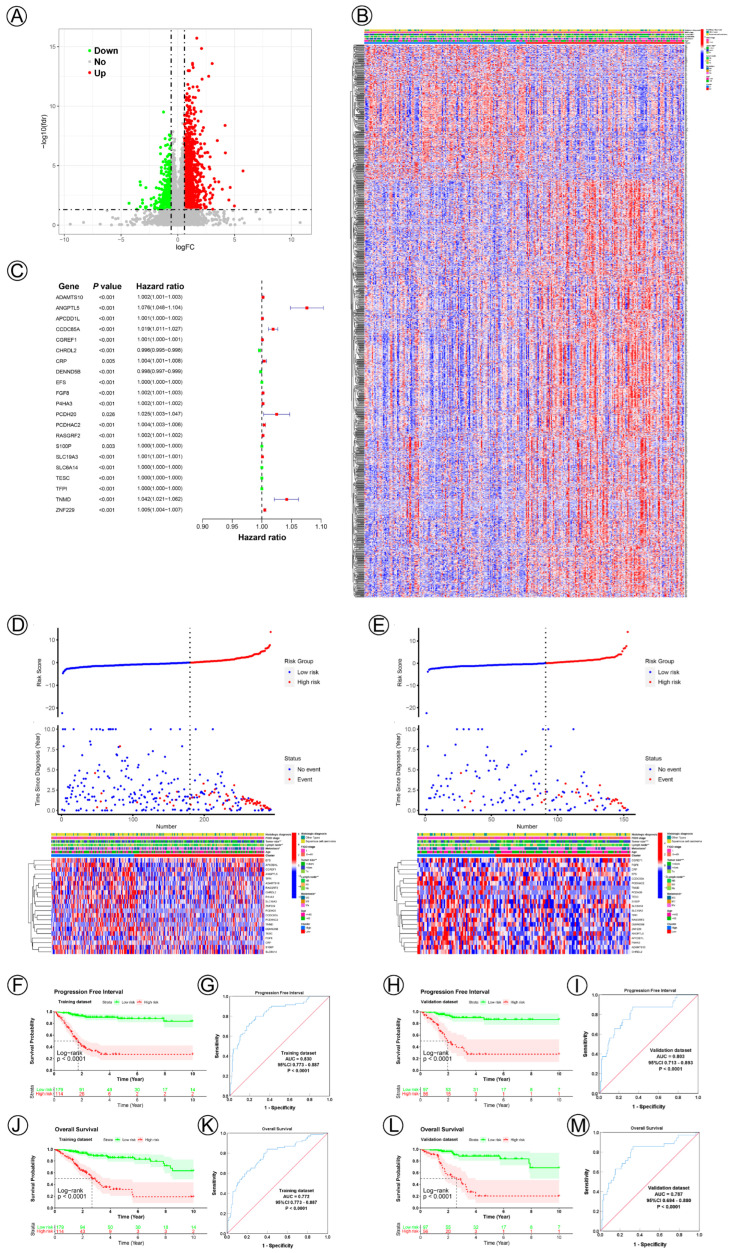
Twenty-one NAD+ metabolic-related gene signatures for predicting the prognosis of cervical cancer. (**A**) A volcano plot of differentially expressed genes between Cluster 1 and Cluster 2. (**B**) A heatmap plot of the expression levels of differentially expressed genes between Cluster 1 and Cluster 2. (**C**) A forest plot of multivariable Cox regression analyses for cervical cancer. (**D**,**E**) The distribution and survival status of cervical cancer patients with different risk scores and the expression levels of the 21 NAD+ metabolic-related genes in training and validation datasets. The blue and red dots represented clinical events or no clinical events. (**F**,**J**,**H**,**L**) KM curves of PFI and OS for high- and low-risk groups in the training and validation datasets. (**G**,**K**,**I**,**M**) ROC analysis showed the sensitivity and specificity of 21 NAD+ metabolic-related gene signatures for predicting PFI and OS for high- and low-risk groups in the training and validation datasets.

**Figure 3 jpm-12-02031-f003:**
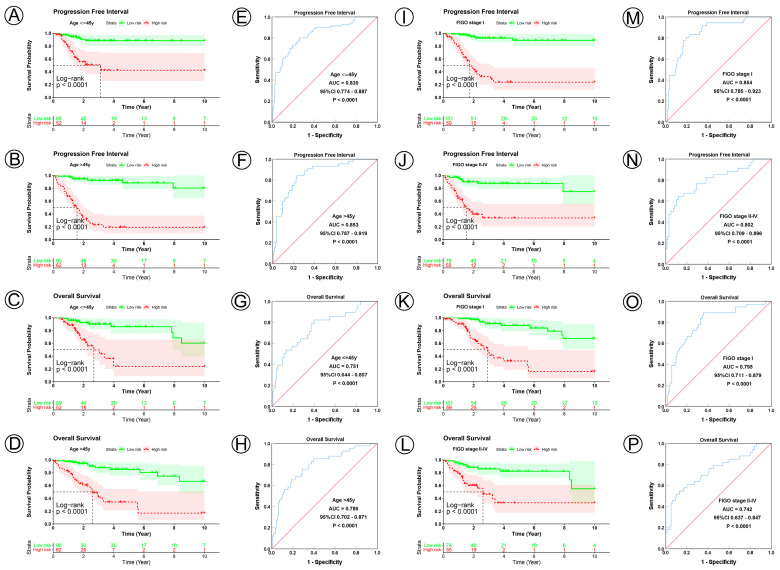
KM and ROC curve analyses of patients stratified by age and FIGO stage. (**A**–**D**) KM curves of PFI and OS for high- and low-risk groups in the ≤45 years and >45 years subgroup. (**E**–**H**) ROC analysis showed the sensitivity and specificity of 21 NAD+ metabolic-related gene signatures for predicting PFI and OS for high- and low-risk groups in the ≤45 years and >45 years subgroup. (**I**–**L**) KM curves of PFI and OS for high- and low-risk groups in the FIGO I stage and FIGO II–IV stage subgroup. (**M**–**P**) ROC analysis showed the sensitivity and specificity of 21 NAD+ metabolic-related gene signatures for predicting PFI and OS for high- and low-risk groups in the FIGO I stage and FIGO II–IV stage subgroup.

**Figure 4 jpm-12-02031-f004:**
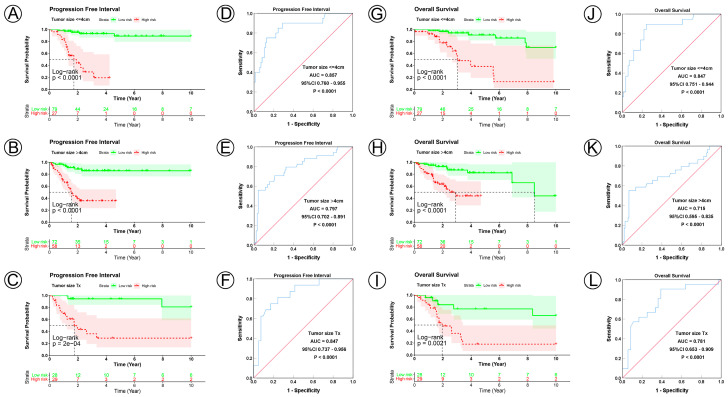
KM and ROC curve analyses of patients stratified by tumor size status. (**A**–**C**,**G**–**I**) KM curves of PFI and OS for high- and low-risk groups in the ≤4 cm, >4 cm and Tx subgroup. (**D**–**F**,**J**–**L**) ROC analysis showed the sensitivity and specificity of 21 NAD+ metabolic-related gene signatures for predicting PFI and OS for high- and low-risk groups in the ≤4 cm, >4 cm and Tx subgroup.

**Figure 5 jpm-12-02031-f005:**
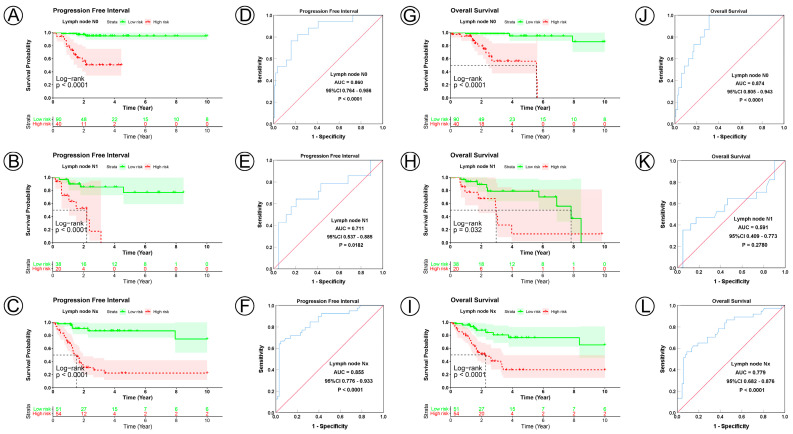
KM and ROC curve analyses of patients stratified by lymph node status. (**A**–**C**,**G**–**I**) KM curves of PFI and OS for high- and low-risk groups in the N0, N1 and Nx subgroup. (**D**–**F**,**J**–**L**) ROC analysis showed the sensitivity and specificity of 21 NAD+ metabolic-related gene signatures for predicting PFI and OS for high- and low-risk groups in the N0, N1 and Nx subgroup.

**Figure 6 jpm-12-02031-f006:**
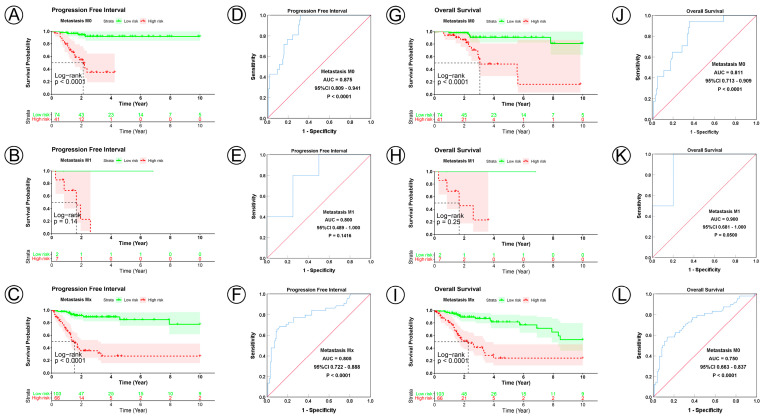
KM and ROC curve analyses of patients stratified by metastasis status. (**A**–**C**,**G**–**I**) KM curves of PFI and OS for high- and low-risk groups in the M0, M1 and Mx subgroup. (**D**–**F**,**J**–**L**) ROC analysis showed the sensitivity and specificity of 21 NAD+ metabolic-related gene signatures for predicting PFI and OS for high- and low-risk groups in the M0, M1 and Mx subgroup.

**Figure 7 jpm-12-02031-f007:**
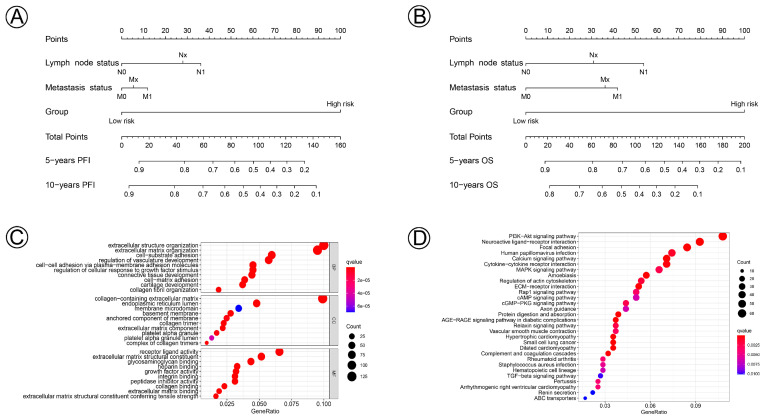
Nomogram for predicting 5- and 10-year PFI and OS of patients with cervical cancer and biological functions of the differential gene expressions in the NAD+ metabolic-related clusters. (**A**,**B**) A nomogram incorporating lymph node status, metastasis status, and risk group was a predictor for 5- and 10-year PFI and OS. (**C**) Gene ontology (GO) annotation (biological process (BP), cellular component (CC), and molecular function (MF)) of DGEs. (**D**) KEGG pathway analysis of DGEs.

**Table 1 jpm-12-02031-t001:** Demographic and Clinical, Pathologic Characteristics of Patients with Cervical Cancer.

Variable	Training Dataset						Validation Dataset			
	Total	Risk Group		χ^2^	*p* Value	Total	Risk Group		χ^2^	*p* Value
		Lower	Higher				Lower	Higher		
	*n* = 293	*n* = 717	*n* =308			*n* = 153	97	56		
Age, y										
≤45	141	89	52	0.470	0.493	75	52	23	2.233	0.135
>45	152	90	62			78	45	33		
FIGO stage										
I	160	101	59	0.613	0.434	90	60	30	1.006	0.316
II–IV	133	78	55			63	37	26		
Tumor size status										
≤4 cm	106	79	27	13.268	0.001	58	49	9	19.358	<0.001
>4 cm	130	72	58			64	35	29		
Tx	57	28	29			31	13	18		
Lymph node status										
N0	130	90	40	11.026	0.004	66	51	15	12.441	0.002
N1	58	38	20			31	20	11		
Nx	105	51	54			56	26	30		
Metastasis status										
M0	115	72	41	6.235	0.044	57	40	17	6.772 ^a^	0.034
M1	9	2	7			6	1	5		
Mx	169	103	66			90	56	34		

^a^ Likelihood Ratio.

**Table 2 jpm-12-02031-t002:** Univariate and Multivariate Cox proportional hazard models of PFI and OS in Cervical Cancer.

Variables		Progression Free Interval							Overall Survival						
		Univariate			Multivariate				Univariate				Multivariate		
	HR	95%CI	*p* Value		HR	95%CI	*p* Value		HR	95%CI	*p* Value		HR	95%CI	*p* Value
Age															
>45 y	1.541	0.949–2.503	0.081						1.230	0.760–1.989	0.400				
FIGO stage															
II–IV	1.255	0.785–2.006	0.343						1.315	0.819–2.112	0.257				
Tumor size															
>4 cm	1.736	0.998–3.018	0.051		0.890	0.479–1.652	0.712		1.828	1.021–3.272	0.042		1.318	0.689–2.520	0.404
Tx	1.788	0.926–3.456	0.084		0.536	0.232–1.237	0.144		2.443	1.309–4.560	0.005		1.097	0.464–2.592	0.834
Lymph node status															
N1	2.174	1.071–4.412	0.032		2.213	1.082–4.526	0.030		2.855	1.424–5.721	0.003		2.505	1.239–5.062	0.011
Nx	3.016	1.706–5.334	<0.001		2.352	1.126–4.916	0.023		3.307	1.814–6.029	<0.001		1.628	0.735–3.607	0.230
Metastasis status															
M1	3.640	1.370–9.674	0.001		1.394	0.486–3.997	0.536		4.163	1.396–12.413	0.011		1.903	0.592–6.114	0.280
Mx	1.550	0.921–2.608	0.099		1.235	0.669–2.279	0.501		2.159	1.241–3.756	0.006		1.871	0.970–3.608	0.062
Risk group															
High risk	10.256	5.647–18.625	<0.001		9.794	5.312–18.056	<0.001		5.980	3.528–10.134	<0.001		5.681	3.298–9.785	<0.001

PFI, Progression Free Interval; CI, Confidence Interval; HR, Hazard Ratio.

**Table 3 jpm-12-02031-t003:** Result of Kaplan-Meier and ROC analysis based on different regrouping methods.

Regrouping Factors	Subgroup	Sample Size	Progression Free Interval				Overall Survival			
			Kaplan-Meier	ROC			Kaplan-Meier	ROC		
			*p* Value	AUC	95%CI	*p* Value	*p* Value	AUC	95%CI	*p* Value
Age, y										
	≤45	141	<0.0001	0.830	0.774–0.887	<0.0001	<0.0001	0.751	0.644–0.857	<0.0001
	>45	152	<0.0001	0.853	0.787–0.919	<0.0001	<0.0001	0.786	0.702–0.871	<0.0001
FIGO stage										
	I	160	<0.0001	0.854	0.785–0.923	<0.0001	<0.0001	0.795	0.711–0.879	<0.0001
	II–IV	133	<0.0001	0.802	0.709–0.896	<0.0001	<0.0001	0.742	0.637–0.847	<0.0001
Tumor size status										
	≤4 cm	106	<0.0001	0.857	0.760–0.955	<0.0001	<0.0001	0.847	0.751–0.944	<0.0001
	>4 cm	130	<0.0001	0.797	0.702–0.891	<0.0001	<0.0001	0.715	0.595–0.835	<0.0001
	Tx	57	0.0002	0.847	0.737–0.956	<0.0001	0.0021	0.781	0.653–0.909	<0.0001
Lymph node status										
	N0	130	<0.0001	0.860	0.764–0.956	<0.0001	<0.0001	0.874	0.805–0.943	<0.0001
	N1	58	<0.0001	0.711	0.537–0.885	0.018	0.0320	0.591	0.409–0.773	0.2780
	Nx	105	<0.0001	0.855	0.776–0.933	<0.0001	<0.0001	0.779	0.682–0.876	<0.0001
Metastasis status										
	M0	115	<0.0001	0.875	0.809–0.941	<0.0001	<0.0001	0.811	0.713–0.909	<0.0001
	M1	9	0.1400	0.800	0.489–1.000	0.1416	0.2500	0.900	0.681–1.00	0.0500
	Mx	169	<0.0001	0.805	0.722–0.888	<0.0001	<0.0001	0.750	0.663–0.837	<0.0001

ROC, Receiver operating characteristic; AUC, Area under the curve.

**Table 4 jpm-12-02031-t004:** Top ten up/down NAD+ metabolic-related genes.

Gene	Mean Expression		logFC	*p* Value	FDR	Expression Regulation	Biological Function	Reference
	Cluster1	Cluster2						
AMIGO2	656.03	2089.59	1.67	4.48E-20	1.93E-16	Up	Malignant Progression	[1,2,3,4]
TGM2	5220.71	22,242.27	2.09	4.43E-19	1.43E-15	Up	Malignant Progression	[5,6]
SAMD4A	431.82	1061.23	1.30	9.76E-18	2.52E-14	Up	Inhibit Angiogenesis	[7]
NT5E	355.48	2900.12	3.03	2.04E-18	2.63E-14	Up	Immunotherapy targeter	[8,9]
ANTXR2	733.58	1809.17	1.30	1.53E-17	3.29E-14	Up	Malignant Progression	[10,11]
PRSS23	2982.48	7634.59	1.36	3.11E-17	5.73E-14	Up	Malignant Progression	[12,13]
ARHGAP29	855.33	1612.31	0.91	7.28E-17	1.04E-13	Up	Malignant Progression	[14,15]
MICAL2	1510.53	3084.34	1.03	6.50E-17	1.04E-13	Up	Malignant Progression	[16,17]
AOX1	42.78	282.14	2.72	4.79E-17	3.09E-13	Up	Tumor Suppressor	[18,19]
HRCT1	27.36	120.47	2.14	4.17E-16	5.38E-13	Up	Component of Membrane	[20]
BCL11A	2624.57	1097.93	−1.26	9.50E-13	3.14E-10	Down	Malignant Progression	[21,22]
NMNAT3	769.70	457.02	−0.75	1.98E-10	2.63E-08	Down	NAD+ Metabolism	[23,24]
EFS	4077.12	2548.44	−0.68	2.50E-10	3.26E-08	Down	Malignant Progression	[25,26]
PRIMA1	1264.97	582.95	−1.12	4.68E-10	5.39E-08	Down	Target for Mutant p53	[27,28]
FAM117B	1611.53	977.46	−0.72	6.03E-10	6.59E-08	Down	Small Vessel Disease	[29,30]
RGMA	2284.31	1224.47	−0.90	2.05E-09	1.73E-07	Down	Malignant Progression	[31,32]
BNIPL	1857.96	880.08	−1.08	2.53E-09	2.06E-07	Down	Malignant Progression	[33,34]
PRKX	5039.73	3275.11	−0.62	2.82E-09	2.23E-07	Down	Malignant Progression	[35,36]
CALML5	13,974.56	4872.76	−1.52	3.80E-09	2.92E-07	Down	Tumor Suppressor	[37]
C3orf58	3176.54	1993.21	−0.67	5.07E-09	3.59E-07	Down	Mesenchymal Differentiation	[38,39]

## Data Availability

The data presented in this study are available on request from the corresponding author.

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
