# Peer review of "Prediction of Cervical Cancer Outcome by Identifying and Validating a NAD+ Metabolism-Derived Gene Signature"

_jpm, 2022, doi:10.3390/jpm12122031_

Round 1

Reviewer 1 Report

Dear authors,

It is of utmost importance to achieve if there are any ethical concerns regarding the use of the data collected in this study ((https://portal.gdc.cancer.gov/). It should be clearly stated.

The text should be reorganized and written in infinitive form.

Please verify the nomenclature of the genes.

The reference list can be updated using references, such as:

https://www.mdpi.com/1420-3049/25/20/4826/htm~

https://www.ncbi.nlm.nih.gov/pmc/articles/PMC7024887/

https://www.annualreviews.org/doi/abs/10.1146/annurev-cancerbio-030518-055905

https://www.nature.com/articles/s41598-021-96038-8

I would like to suggest emphasizing the differences between normal vs. tumoral specimens.

Reviewer 2 Report

The authors sought to identify gene signatures to predict cervical cancer progression. They identified NAD+ metabolism-related genes as a predictor of high risk of cervical. Specific comments about the manuscript are listed below.

(1) A major concern with the manuscript is that none of the genes in the set of genes that showed good prognostic values are directly related to NAD+ metabolism. The authors need to discuss how these genes are related to NAD+ metabolism.

(2) The panels E-G were not included in the Figure 7 but they were discussed in the figure legend.

Reviewer 3 Report

In the manuscript entitled “Prediction of cervical cancer outcome by identifying and validating a NAD+ Metabolism-Derived Genes signature”, Chen and coworkers investigated the expression of NAD+-related genes in cervical cancer using available dataset containing gene expression data as well as clinical and demographic  information. The authors identified a specific NAD+ derived gene signatures associated with poor prognosis. The topic presented in this manuscript is interesting and expand our knowledge on the role of NAD+ metabolism in cancer. The results support the conclusion stated by the authors but the manuscript is overall very descriptive and the discussion of the data and implications is somewhat superficial. I would address this main issue prior publication. Additionally, I suggest the following improvements:

Major:

·       Are the genes reported in Fig1A differentially regulated in cervical cancer regardless of the prognosis? Some of these genes were discussed in the discussion sections but not all of them. What do we know about their involvement in other types of cancer?

·       What is the role of the main genes dysregulated in cervical cancer and reported in Table 4? It would be interesting to have a column in the table to add this info and maybe some literature ref? This can also be added to the discussion section

·       A recent paper reported the great potential of NAD+ related genes as predictors for ovarian cancer prognosis and immunotherapy response (doi: 10.3389/fgene.2022.905238).  I think this refence should be included and discussed in the manuscript.

·        

Minor:

·       Each acronym should be defined in the text only one time, for example NAD+ in the introduction.

·       English language check is required thought the manuscript to improve clarity. An example is the sentence in the abstract: “Excellent clinical outcomes achieved with current screening test and medical treatment in early stages, while advanced stage with poor prognosis.” which is incorrect and unclear.
